# Amygdala EFP Neurofeedback Effects on PTSD Symptom Clusters and Emotional Regulation Processes

**DOI:** 10.3390/jcm14072421

**Published:** 2025-04-02

**Authors:** Nadav Goldental, Raz Gross, Daniela Amital, Eiran V. Harel, Talma Hendler, Aron Tendler, Liora Levi, Dmitri Lavro, Tal Harmelech, Shulamit Grinapol, Nitsa Nacasch, Eyal Fruchter

**Affiliations:** 1Division of Psychiatry, Chaim Sheba Medical Center, Tel Hashomer 52621, Israel; nadav.goldental@sheba.health.gov.il; 2Department of Epidemiology, School of Public Health and Department of Psychiatry, School of Medicine, Tel Aviv University, Sheba Medical Center, Tel Aviv 6997801, Israel; raz.gross@sheba.health.gov.il; 3Division of Psychiatry, Barzilai Medical Center, Ashkelon 7830604, Israel; danielaam@bmc.gov.il; 4Be’er Ya’akov Mental Health Center, Be’er Ya’akov 70350, Israel; eiranharel@gmail.com; 5Sagol Brain Institute, Tel Aviv Sourasky Medical Center, School of Psychological Sciences, Faculty of Medical and Health Sciences and Sagol School of Neuroscience, Tel-Aviv 6997801, Israel; hendlert@gmail.com; 6GrayMatters Health Ltd., Haifa 3303403, Israel; liora@graymatters-health.com (L.L.); dima@graymatters-health.com (D.L.);; 7Department of Community Mental Health, University of Haifa, Haifa 3498838, Israel; s.grinapol@rambam.health.gov.il; 8Clalit Health Services Community Division, Ramat-Chen Brull Mental Health Center, Tel Aviv-Yafo 6719709, Israel; nitsana@sheba.health.gov.il; 9ICAR Collective and the Brus Rappaport Medical Faculty of the Technion, Haifa 3200003, Israel; eyal.fruchter@psmh.health.gov.il

**Keywords:** self-neuromodulation, EFP-neurofeedback, amygdala downregulation, emotion regulation

## Abstract

**Background:** Post-traumatic stress disorder (PTSD) manifests through distinct symptom clusters that can respond differently to treatments. Neurofeedback guided by the Amygdala-derived-EEG-fMRI-Pattern (Amyg-EFP-NF) has been utilized to train PTSD patients to regulate amygdala-related activity and decrease symptoms. **Methods:** We conducted a combined analysis of 128 PTSD patients from three clinical trials of Amyg-EFP-NF to evaluate effects across symptom clusters (as assessed by CAPS-5 subscales) and on emotion regulation processing (evaluated by the ERQ). **Results:** Amyg-EFP-NF significantly reduced severity across all PTSD symptom clusters immediately post-treatment, with improvements maintained at three-month follow-up. The arousal and reactivity cluster showed continued significant improvement during follow-up. Combined effect sizes were large (η^2^_p_ = 0.23–0.35) across all symptom clusters. Regression analysis revealed that emotion regulation processes significantly explained 17% of the variance in symptom improvement during the follow-up period. **Conclusions:** Reduction of PTSD symptoms following Amyg-EFP-NF occurs across all symptom clusters, with emotional regulation processes potentially serving as an underlying mechanism of action. These results support Amyg-EFP-NF as a comprehensive treatment approach for PTSD that continues to show benefits after treatment completion.

## 1. Introduction

Post-traumatic stress disorder (PTSD) affects 5–9% of individuals exposed to traumatic events, with over 70% of adults worldwide experiencing potentially traumatic events during their lifetime [1]. PTSD is a multifaceted, highly heterogeneous syndrome that can theoretically manifest in many different combinations of symptoms [2]. The DSM-5 [3] identifies four primary symptom clusters that are central to PTSD diagnosis and treatment: intrusive reexperiencing of traumatic events (Cluster B), avoidance of external or internal trauma-related stimuli (Cluster C), negative alterations in cognition and mood (Cluster D), and arousal and reactivity changes (Cluster E). This categorization provides a framework for characterizing the disorder in light of mental processes underlying treatment mechanisms [4,5].

The clinician-administered PTSD scale for DSM-5 (CAPS-5), the leading assessment tool of PTSD symptom severity, provides separate scores for each symptom cluster [6,7], enabling comprehensive clinical characterization and detailed tracking of treatment response [7]. This granular assessment approach advances precision medicine by allowing treatment personalization based on individual symptom patterns [8,9]. Understanding how treatments affect specific symptom clusters requires identifying the underlying mental processes that link these diverse manifestations.

Emotion regulation emerges as a critical mechanism potentially bridging these diverse symptom manifestations. Defined as the ability to monitor, identify, and modify emotional responses and/or behavioral manifestations [10], emotion regulation represents a promising candidate for understanding the complex neurobiological processes underlying PTSD. Research distinguishes between automatic-affect based processes (implicit emotion regulation) and elaborated cognitive-based processes (explicit emotion regulation) [11]. This process-based approach aligns with the Research of Domain Criteria (RDoC) framework, which integrates neurobiology, psychology, and genetic findings to understand psychiatric disorders [8,9,12]. The impairment of emotion regulation in PTSD is well-documented, with patients often showing excessive reliance on maladaptive strategies, like emotional suppression and avoidance, while struggling with adaptive strategies such as cognitive reappraisal [13]. Given emotion regulation’s central role in PTSD, the amygdala stands out as a critical therapeutic target, serving as a key structure in both emotion regulation and PTSD pathophysiology [13]. Beyond its established role in fear and stress processing, the amygdala serves as a crucial node in emotion regulation networks ([14,15,16,17]). It contributes to both implicit and explicit regulation through distinct connectivity patterns with the ventromedial and dorsolateral prefrontal cortex, respectively [18]. Despite this central role, amygdala activity is not directly targeted by current non-invasive treatments for PTSD, except with self-neuromodulation techniques, known as neurofeedback (NF).

While several NF approaches exist for treating psychiatric conditions, Amygdala-derived EEG-fMRI-Pattern neurofeedback (Amyg-EFP-NF) offers distinct advantages over traditional techniques [14]. Unlike conventional EEG-NF that targets cortical activity patterns, Amyg-EFP-NF specifically monitors deep limbic activity through its unique algorithm derived from simultaneous EEG-fMRI recordings. This provides the spatial precision of fMRI (targeting the amygdala) with the temporal resolution and accessibility of EEG. Alternative approaches like conventional EEG-NF lack anatomical specificity for subcortical structures, while pure fMRI-NF faces practical limitations, including cost, accessibility, and temporal resolution, that restrict clinical implementation [15]. The Amyg-EFP technology bridges this gap by leveraging machine learning to extract amygdala-specific signals from scalp EEG, enabling targeted modulation of a key structure in PTSD pathophysiology in an accessible clinical format.

While functional Magnetic Resonance Imaging-based NF (fMRI-NF) enables precise anatomical targeting of the amygdala and has shown promise in PTSD treatment [16,17,19,20], its limited temporal resolution and practical constraints reduce clinical applicability. Conversely, electroencephalography (EEG) offers superior temporal resolution and scalability (access and cost) [21] but lacks precise localization of deep brain activity [22]. An innovative solution combines these approaches through machine learning applied to simultaneous EEG-fMRI data, generating an Amygdala-derived-EEG-fMRI-Pattern (Amyg-EFP) for continuous, real-time monitoring of amygdala activation [23,24,25]. This technology provides feedback through an interactive audio-visual display that changes with amygdala-derived EFP signal modulation (hereby termed Amyg-EFP-NF) [26].

Initial studies of Amyg-EFP-NF in PTSD have shown promising results [27,28,29]. While all studies demonstrated significant reductions in the total score of the CAPS-5 immediately post-treatment and on the patient-rated PTSD checklist at both post-treatment and at three-month follow-up, their results regarding the CAPS-5 subscales varied. Fine et al. [28] reported consistent reduction across subscales, Fruchtman et al. [27] showed varied results across treatment arms, and Fruchter et al.’s [29] larger study did not report subscale findings.

The current study combines data from these three trials (N = 128) to comprehensively evaluate Amyg-EFP-NF’s effectiveness across PTSD symptom clusters and examine the role of emotion regulation in treatment outcomes. By targeting amygdala activity and potentially restructuring emotion regulation neural circuits, we hypothesized that: (1) Amyg-EFP-NF would produce significant improvements across all symptom clusters as measured by CAPS-5 subscales, (2) these improvements would maintain or increase at three-month follow-up, and (3) changes in emotion regulation would be associated with changes in symptom clusters.

## 2. Materials and Methods

**Population:** A total of 128 patients (64.06% females and 35.94% males) with a mean age (±SD) of 38.26 (±11.19) were available for the data re-analysis. The unbalanced gender reflects the different composition and diagnosis in each study. Study 1 included males and females diagnosed with chronic PTSD, Study 2 included only females diagnosed with complex PTSD due to childhood sexual abuse, and Study 3 included males and females diagnosed with chronic PTSD. Therefore, on the whole, there were more females in the complete analysis cohort.

Table 1 presents detailed demographic and clinical characteristics across the three studies. Inclusion criteria across studies required participants to be 18–65 years old with a confirmed PTSD diagnosis according to DSM-5 criteria and a minimum CAPS-5 score of 25. Exclusion criteria included severe psychiatric comorbidity (psychosis, bipolar disorder, active substance use disorder), moderate to severe traumatic brain injury, ongoing psychotherapy initiated within the past 3 months, and change in psychiatric medication within 4 weeks of screening. Study 2 specifically recruited females with complex PTSD related to childhood sexual abuse, while Studies 1 and 3 included participants with various trauma types (combat, interpersonal violence, accidents, and terror attacks). Participants in Study 2 were also required to be currently engaged in psychotherapy.

**NF training design:** The detailed methods and study design for the studies included in the this re-analysis were previously published in Fruchtman et al. [26] (Study 1), Fine et al. [27] (Study 2), and Fruchter et al. [28] (Study 3). In short, Studies 1 and 2 assessed the clinical effect of an Amyg-EFP-NF investigational device in chronic PTSD and childhood sexual abuse survivors with treatment-refractory PTSD (also regarded as complex-PTSD), respectively. Study 3 evaluated the safety and efficacy of Prism™, an FDA-approved amygdala-derived-EFP-NF system for the treatment of chronic PTSD populations. **Study 1** employed 15 sessions of Amyg-EFP-NF via a three-arm randomized controlled trial. Both active arms used a neutral feedback interface of a waiting room (Neutral-NF), but one arm also incorporated, from the sixth session onward, an individually tailored trauma-focus script as an auditory feedback with volume corresponding to the Amyg-EFP modulation. **Study 2** Employed 10 sessions of Amyg-EFP-NF in a randomized controlled manner, obtained either NF with intensive psychotherapy or intensive psychotherapy alone. **Study 3** employed 15 NF sessions over 8 weeks, using the commercial version of Amyg-EFP-NF of the Prism software.

To maintain NF training quality across studies, standardized protocols were implemented for each trial. All studies utilized the same core AmygEFP technology, with consistent EEG acquisition parameters using the V-Amp™ EEG amplifier and BrainCap™ electrode positioning according to the standard 10/20 system. The AmygEFP signal calculation methodology was identical across studies, employing the same machine learning algorithm for predicting amygdala BOLD activity from EEG data. Training protocols shared similar session structures featuring baseline, regulation, and debriefing periods, though they varied in the number of sessions (10–15) and specific feedback interfaces. Each study adhered to standardized participant instructions regarding mental strategy exploration during the regulation periods. This consistency in technical implementation and procedural structure ensured comparable NF experiences despite differences in patient populations and study settings

**Outcome measures:** In all studies, the primary outcome was assessed at baseline (Baseline), immediately post-treatment (Immediate-Post), and at three months post-treatment (3 M Follow-up). The primary endpoint in all studies was the proportion of subjects demonstrating a clinically meaningful improvement in CAPS-5 score or PTSD checklist (PCL) from Baseline to the 3 M Follow-up visit. Studies 1 and 2 only measured the PCL at 3 M follow-up. Study 3 measured the CAPS-5 at 3 M Follow-up as well. All three studies recorded the emotion regulation questionnaire (ERQ) at Baseline, Immediate-Post, and at 3 M Follow-up.

The CAPS-5 is the gold standard clinician-administered interview for PTSD diagnosis and severity assessment. It demonstrates excellent psychometric properties with internal consistency (Cronbach’s α = 0.88–0.94), inter-rater reliability (ICC = 0.78–0.83), and test-retest reliability (r = 0.83) [30]. The CAPS-5 yields a total severity score (0–80) and individual cluster scores: Intrusion (0–20), Avoidance (0–8), Cognition/Mood (0–28), and Arousal/Reactivity (0–24). The ERQ is a 10-item self-report measure assessing two emotion regulation strategies: Cognitive Reappraisal (CR; 6 items) and Expressive Suppression (ES; 4 items). It shows good internal consistency (Cronbach’s α = 0.75–0.82) and test-retest reliability (r = 0.69) [31]. Higher scores on CR indicate greater use of adaptive reinterpretation strategies, while higher ES scores reflect a greater tendency to inhibit emotional expression.

**Data analysis:** In the primary analysis, we integrated the CAPS-5 symptom cluster data (Criteria B-E) from the active treatment groups in Studies 1 and 2, as well as the entire efficacy analysis set of the open-label Study 3. To test the primary outcome of the Amyg-EFP-NF treatment on symptom clusters, we performed a set of linear mixed model (LMM) analyses in R software (version 4.2.1) using the *lme4* package [32]. For each cluster, we included the cluster score as the dependent variable and tested the fixed effect of treatment timepoint (Baseline, Immediate-Post, and 3 M Follow-up). We included an effect of Study and a nested effect of Subjects within Study as random effects. Significance was calculated using the *lmerTest* package [33], which applies Satterthwaite’s method to estimate degrees of freedom and generate *p*-values for mixed models.

Linear mixed models (LMM) were specifically chosen for this combined analysis because of their robust capabilities in handling multi-level data structures and longitudinal designs with missing observations. LMM offers significant advantages over traditional repeated measures ANOVA for several reasons particularly relevant to our dataset. First, the nested structure of participants within studies creates a hierarchical data structure that LMM handles naturally by accounting for within-study correlations. Second, LMM accommodates the unbalanced design resulting from the varying follow-up protocols across studies (with only Study 3 including 3-month CAPS-5 follow-up assessments), maximizing statistical power by utilizing all available data points without requiring complete cases. Third, LMM allows for the explicit modeling of random effects at both the study and subject levels, correctly attributing variance to different sources and reducing the risk of Type I errors. Finally, using Satterthwaite’s method to estimate degrees of freedom provides more accurate *p*-values when sample sizes differ across groups and timepoints. This statistical approach effectively addresses the heterogeneity across the three component studies while maintaining statistical rigor in the combined analysis.

Post-hoc comparisons were performed using the Tukey HSD test, and effect sizes for the model were computed using the *effect size* package [34]. The LMM approach was well-suited for our pooled analysis design for two key reasons. Firstly, in our design, the subjects from each study could be accurately defined as nested within the study as a random factor. Secondly, the combination of the three studies is not fully factorial, meaning not all subjects completed the 3-month follow-up. Despite this, the LMM approach effectively handles the missing observations and makes the best use of the available data from the combined studies. For the secondary exploratory analysis, we utilized change scores obtained from the ERQ, a widely used questionnaire for assessing Cognitive Reappraisal (CR) and Expressive Suppression (ES) components of regulation processing [35], along with change score in total CAPS-5. Change scores were calculated for Baseline versus Immediate-Post and for Immediate-Post versus 3 M Follow-up. For each change score, we used multiple regression analysis to determine if changes in the ERQ components could explain the observed differences in total CAPS-5.

## 3. Results

Table 2 shows the descriptive statistics of CAPS-5 cluster scores across different timepoints and studies. Note that 3 M Follow-up was only available for Study 3. All clusters showed a significant effect for timepoint for Baseline vs. Immediate-Post and for Baseline vs 3 M Follow-up. For **Cluster B** (Figure 1a), the mixed-model analysis showed a significant effect of timepoint (*F* (2, 182.82) = 49.12, *p* < 0.001) with a large effect size (η^2^_p_ = 0.35, 95% CI [0.24, 0.44]). The fixed effects of timepoint further indicated significant differences between Baseline and Immediate-Post (*b* = −2.66, SE = 0.31, *t* (187.88) = −8.60, *p* < 0.001), as well as between Baseline and 3 M Follow-up (*b* = −3.18, SE = 0.41, *t* (178.16) = −7.84, *p* < 0.001). The post-hoc contrast testing the difference between Immediate-Post and 3 M Follow-up was not significant (*b* = −0.51, SE = 0.42, *t* (180) = −1.22, *p* = 0.445). For **Cluster C** (Figure 1b), the mixed-model analysis also showed a significant effect of Timepoint (*F* (2, 202.08) = 30.40, *p* < 0.001) with a large effect size (η^2^_p_ = 0.23, 95% CI [0.13, 0.32]). The fixed effects of timepoint further indicated significant differences between Baseline and Immediate-Post (*b* = −1.12, SE = 0.17, *t* (191.65) = −6.43, *p* < 0.001), as well as between Baseline and 3 M Follow-up (*b* = −1.48, SE = 0.23, *t* (210.05) = −6.55, *p* < 0.001). The post-hoc contrast testing the difference between Immediate-Post and 3 M Follow-up was not significant (*b* = −0.36, SE = 0.24, *t* (159) = −1.50, *p* = 0.293). For **Cluster D** (Figure 1c), the mixed-model analysis similarly found a significant effect of timepoint (*F* (2, 192.31) = 42.46, *p* < 0.001) with a large effect size (η^2^_p_ = 0.31, 95% CI [0.20, 0.40]). The fixed effects indicated significant differences between Baseline and Immediate-Post (*b* = −3.88, SE = 0.46, *t* (190.77) = −8.41, *p* < 0.001), as well as between Baseline and 3 M Follow-up (*b* = −4.08, SE = 0.61, *t* (193.42) = −6.66, *p* < 0.001). The post-hoc contrast testing the difference between Immediate-Post and 3 M Follow-up was not significant (*b* = −0.19, SE = 0.62, *t* (192) < 1). Lastly, for **Cluster E** (Figure 1d), the mixed model analysis showed a significant effect of timepoint (*F* (2, 184.52) = 37.30, *p* < 0.001) with a large effect size (η^2^_p_ = 0.29, 95% CI [0.18, 0.38]). The fixed effects of timepoint indicated significant differences between Baseline and Immediate-Post (*b* = −2.33, SE = 0.35, *t* (190.82) = −6.57, *p* < 0.001), as well as between Baseline and 3 M Follow-up (*b* = −3.56, SE = 0.46, *t* (178.79) = −7.70, *p* < 0.001). In contrast to all other clusters, the post-hoc contrast testing the difference between Immediate-Post and 3 M Follow-up was also significant (*b* = −1.23, SE = 0.49, *t* (173) < −2.54, *p* = 0.032), indicating an additional drop in arousal and reactivity in the later phase following the end of treatment.

The analysis demonstrated significant effects across all symptom clusters with large effect sizes (η^2^_p_ ranging from 0.23 to 0.35). These effect sizes, representing the proportion of variance explained by the treatment across the three timepoints (Baseline, Immediate-Post, and 3 M Follow-up), indicate robust treatment effects comparable to Cohen’s d > 0.8 or correlations > 0.5.

To provide a better understanding of the cross-cluster improvement in CAPS-5, we leveraged additional data from *Study 3* (chronic PTSD) on the ERQ. Specifically, we tested whether the Cognitive Reappraisal and Expressive Suppression subcomponents of the ERQ could explain the observed changes in the total CAPS-5 scores between Baseline, Immediate-Post, and 3 M Follow-up. Table 3 outlines the mean differences across measurement timepoints, indicating the anticipated clinical effects of treatment on the examined scores. The data show that alongside the decrease in CAPS-5 total scores, there is a slight reduction in Expressive Suppression and a slight enhancement in Cognitive Reappraisal after treatment. This suggests a diminished tendency to inhibit outward emotional expression and an improved ability to positively reinterpret or reframe emotional experiences, respectively. For a detailed analysis of the mean differences in CAPS-5 and ERQ between measurement timepoints, please refer to Fruchter et al. [29]. To further analyze the association between CAPS-5 and emotional regulation measurements, we employed a regression analysis with CAPS-5 as the dependent variable and ERQ subscales as independent variables.

Separate models were employed for each timepoint period (Baseline to Immediate-Post, Baseline to 3 M Follow-up, and Immediate-Post to 3 M Follow-up). To account for the separate periods that were tested in each model, we used adjusted *p*-values with the Bonferroni method for multiple comparisons. For the period between Baseline and Immediate-Post (*R*^2^ = 0.070, *F* (2, 60) = 2.263, *p* = 0.339) (Figure 2a,b) and the period between Baseline to 3 M Follow-up (*R*^2^ = 0.080, *F* (2, 60) = 2.605, *p* = 0.246), it did not yield significant results, indicating that the ERQ components could not explain the difference in CAPS-5 in these periods (Figure 2c,d). As noted, Studies 1 and 3 did not include 3 M Follow-up data for CAPS-5 and ERQ in their protocols. Nevertheless, if we include the available data from Study 1 (19 out of 25 subjects) and Study 2 (27 out of 37 subjects) in a pairwise manner for the Baseline to Immediate-Post period (109 subjects overall), the regression analysis for this period remains unchanged and not significant (*R*^2^ = 0.025, *F* (2, 106) = 1.374, *p* = 0.257). In contrast, for the period from Immediate-Post to 3 M Follow-up, the ERQ components explained 17% of the variance in CAPS-5 change (Total Effect: *R*^2^ = 0.166, *F* (2, 60) = 5.963, *p* = 0.012; ES contribution: β = 0.653, *p* = 0.006, *r*_partial_ = 0.344; CR contribution: β = −0.348, *p* = 0.039, *r*_partial_ = −0.263), suggesting a significant contribution of emotion regulation to this incremental long-term improvement in PTSD symptoms (Figure 2e,f).

## 4. Discussion

The present study demonstrates the broad therapeutic potential of amygdala-derived-EFP-NF (Prism) for treating PTSD through a comprehensive analysis of symptom cluster responses across three clinical trials. By examining pooled data from both chronic PTSD and complex PTSD populations, we found robust improvement across all DSM-5 symptom clusters immediately following treatment and at three-month follow-up, with overall large effect sizes across both intervals, ranging from 0.23 to 0.35. While all clusters showed sustained improvement at three-month follow-up, arousal and reactivity symptoms continued to show significant incremental improvement from post-treatment to three-month follow-up. Notably, the association between improved emotion regulation capacities and symptom reduction emerged specifically during this follow-up period, explaining 17% of the variance in CAPS-5 improvement. This temporal pattern suggests distinct but complementary mechanisms operating across different timeframes.

Our findings align with and extend recent meta-analytic evidence on evidence-based psychotherapies for PTSD [36]. While both studies demonstrate improvement across all symptom clusters immediately post-treatment, our results uniquely show that these improvements are maintained at three-month follow-up, with Cluster E (arousal/reactivity) showing additional enhancement during this period. This differential pattern in long-term outcomes, particularly for arousal symptoms, suggests potential mechanistic differences in how various symptom clusters respond to neurofeedback over time.

### 4.1. Robustness and Long-Term Effects of Treatment

The convergence of results provides compelling evidence for treatment robustness, demonstrating consistent long-term improvement across diverse populations and study designs. This sustained therapeutic effect aligns with previous findings in NF interventions across various clinical conditions, with similar persistence of clinical benefits documented in fMRI-NF studies of obsessive-compulsive disorder and Tourette’s [37], fibromyalgia [38], and nicotine addiction [39].

The persistence of therapeutic effects likely involves multiple complementary mechanisms. At a neural level, repeated NF training may induce lasting neuroplastic changes through the strengthening of pathways supporting adaptive emotional processing. Animal models have demonstrated that repeated training of regulatory pathways strengthens inhibitory connections, particularly between prefrontal regions and the amygdala [40]. Recent neuroimaging research in humans suggests that neurofeedback training induces lasting changes in functional connectivity patterns that persist well beyond the training period. Additionally, the improvement in symptoms might create a positive feedback loop: as patients experience better emotional control, they engage more confidently in previously avoided situations, providing natural opportunities to further strengthen their regulatory capabilities. This “practice effect” in real-world settings could explain why some benefits continue to emerge during the follow-up period, such as improvement in arousal and reactivity symptoms (Cluster E; Figure 1d). Such a gradual strengthening of treatment effects aligns with learning theory perspectives on NF, where initial explicit learning transitions to implicit mastery—similar to how other complex skills are acquired [21,26]. Future work should assess the real-world practice of successful mental strategies and the neural changes following treatment and in follow-up to better understand these mechanisms.

### 4.2. Cross-Cluster Symptom Improvement

The uniform improvement across symptom clusters represents a notable departure from traditional PTSD treatments that often show differential effects. Current pharmacological approaches typically demonstrate selective efficacy: SSRIs and SNRIs show stronger effects on intrusion and avoidance symptoms, while atypical antipsychotics primarily target arousal/reactivity symptoms [41]. This differential response pattern in conventional treatments reflects their specific molecular targets—serotonergic systems primarily affecting emotional memory processing and threat response versus dopaminergic systems modulating arousal and attention [42,43].

Recent research has illuminated the complex interrelationships between PTSD symptom clusters and their broader impact on health outcomes. For example, the negative alterations in cognition and mood (Cluster D) have emerged as a significant predictor of suicidal ideation among veterans [44], highlighting the contribution to the disorder’s threatening debilitation. Similarly, arousal/reactivity symptoms (Cluster E) demonstrate unique contributions to cardiovascular reactivity during stress in trauma-exposed individuals [45], emphasizing the disturbance of salience and vigilance processing. Finally, the intrusion symptoms (Cluster B) and avoidance symptoms (Cluster C) similarly reflect different manifestations of disrupted emotion-cognition integration of memory and motivation [46,47]. Rather than treating these clusters as separate phenomena requiring distinct interventions, Amyg-EFP-NF appears to address their common underlying mechanism, possibly through the acquired regulation of shared neural function [48,49].

### 4.3. Neural Circuit Mechanisms

The comprehensive clinical improvement observed with Amyg-EFP-NF suggests it targets fundamental neural processes underlying various PTSD manifestations. Recent mechanistic research has revealed that rather than simply exhibiting heightened activity, PTSD involves complex disruptions in amygdala-derived networks that support both emotion generation and regulation [50,51]. The amygdala contains distinct neuronal populations mediating both threat detection and safety learning [52], with separate circuits supporting fear acquisition versus fear extinction [53]. Rather than just showing heightened activity, PTSD involves an imbalance between these opposing circuits, with hyperactive threat-detection systems overwhelming safety-learning mechanisms [54].

This neurobiological targeting is particularly notable when compared with other emerging digital health interventions for PTSD. While digital mental health interventions have shown moderate efficacy for reducing PTSD symptoms [55], they typically do not directly target these underlying neural mechanisms. Similarly, virtual reality exposure therapies, though promising for addressing avoidance symptoms, have shown variable effects across symptom clusters [56]. In contrast, Amyg-EFP-NF’s consistent improvement across all symptom dimensions suggests advantages of directly engaging the neural circuitry implicated in diverse symptom manifestations. However, comparative effectiveness studies are needed to directly assess the relative benefits of these approaches and identify optimal candidates for each intervention type.

Through real-time feedback, patients learn to selectively engage safety-promoting circuits while dampening threat-responsive networks. The amygdala serves as a critical hub in an extended network that includes the ventromedial prefrontal cortex for implicit emotion regulation, dorsolateral prefrontal regions for explicit control, and brainstem areas for autonomic regulation [18,32]. While each of these regions shows altered activity patterns in PTSD, the coordination between regions appears most disrupted. By targeting the amygdala as a key node for this coordination, the neurofeedback may facilitate the re-establishment of healthy network dynamics across these circuits.

The persistence of improvement in arousal/reactivity symptoms during follow-up likely reflects the gradual strengthening of these regulatory circuits. The amygdala’s reciprocal connections with brainstem arousal systems and autonomic control regions make it particularly well-positioned to influence these symptoms [33]. As patients develop stronger regulatory capabilities through NF training, they may become increasingly able to modulate these circuits in daily life, leading to progressive improvement in arousal-related symptoms. This capacity for circuit-level modulation aligns with the conceptualization of Amygdala-NF as an emotion regulation training procedure [34,35,51], where patients learn to regulate their emotional responses through direct feedback about amygdala activity patterns.

### 4.4. Emotion Regulation Mechanisms

The temporal pattern of improvement in emotion regulation provides important insight into the treatment’s mechanism of action. While symptom reduction begins during active treatment, the association between improved emotion regulation capacities and continued symptom reduction emerges during the follow-up period, suggesting that Amyg-EFP-NF initiates a cascade of adaptive changes that continue to evolve [51].

A notable finding was the continued improvement in arousal and reactivity symptoms (Cluster E) during the follow-up period, while other clusters maintained their initial gains. This pattern differs somewhat from previous evidence-based psychotherapy studies, which typically show uniform maintenance patterns across clusters [36]. The selective enhancement of arousal regulation may reflect the gradual strengthening of automatic regulatory capabilities through neurofeedback training, suggesting that amygdala-based interventions might have particular utility for the long-term management of hyperarousal symptoms.

Recent work has shown that successful emotion regulation involves flexibly selecting between different strategies based on context—particularly between rapid, automatic regulation for high-intensity emotional situations and more elaborate cognitive strategies for lower-intensity scenarios [57,58]. The NF training appears to help patients transition from over-reliance on automatic suppression toward greater flexibility in strategy selection. This is particularly significant given that PTSD patients often show reduced regulatory flexibility manifested in diminished ability to choose engagement versus disengagement strategies according to situational demands [58].

The differential impact of regulation strategies in PTSD is well-documented: increased use of expressive suppression is associated with worse outcomes and tends to increase physiological arousal and maintain threat associations, while enhanced use of cognitive reappraisal predicts symptom improvement and reduces both subjective distress and amygdala reactivity [59,60]. This pattern is especially relevant given that PTSD patients often default to suppression due to the overwhelming nature of trauma-related emotions [61].

This shift in regulatory capacity involves distinct neural mechanisms—implicit regulation appears to be mediated by ventromedial prefrontal-amygdala circuits, while explicit regulation engages dorsolateral prefrontal regions [18]. Brain imaging work suggests the amygdala serves as a crucial switch point between these regulatory pathways, with its pattern of network connectivity predicting regulatory strategy selection [50]. The delayed emergence of emotion regulation improvements may reflect the time needed to strengthen these complementary neural systems and establish new regulatory habits.

While we observed associations between emotion regulation and symptom improvement, the correlational nature of these findings precludes causal interpretations. Future studies should examine how individual differences in pre-treatment network organization might predict treatment response, particularly focusing on measures of regulatory flexibility and network integration that could help optimize the intervention for different patient profiles [51]. Additionally, investigating the neural mechanisms underlying the transition from immediate symptom reduction to enhanced regulatory capabilities during follow-up could provide insights for improving treatment protocols, including examination of how patients transition between different regulation strategies before, during, and after NF training [58].

### 4.5. Clinical Implications and Future Directions

These findings suggest several important modifications to clinical protocols. Treatment monitoring should extend well beyond the active treatment period to capture full therapeutic benefits, and protocols may benefit from incorporating targeted emotion regulation skills training. The treatment appears particularly valuable for patients with treatment-resistant PTSD, with potential for protocol customization based on individual emotion regulation profiles and predominant symptom patterns. Significant opportunities exist for integration with trauma-focused psychotherapy and emotion regulation skills training.

Several limitations warrant consideration in interpreting these results. The study populations consisted of individuals sufficiently motivated to participate in clinical trials, potentially limiting generalizability. Only Study 3 included CAPS-5 data at three-month follow-up, limiting our ability to examine longer-term effects across the full sample based on this scale. The varying treatment protocols across studies, while supporting robustness, also make it challenging to identify optimal treatment parameters.

Future research should address the optimization of treatment protocols through several approaches. First, follow-up assessments should extend well beyond the three-month period examined here, ideally to 12 months or longer, to fully characterize the durability of treatment effects. Second, incorporating neuroimaging before and after treatment would help elucidate whether Amyg-EFP-NF induces measurable changes in brain structure and connectivity, particularly in circuits connecting the amygdala with prefrontal regulatory regions. Such mechanistic insights could facilitate protocol optimization and potentially help identify neurobiological predictors of treatment response.

Studies examining potential synergies with other therapeutic approaches would enhance the understanding of treatment effectiveness, particularly in identifying how Amyg-EFP-NF might complement existing evidence-based interventions. Investigation of the specific neural mechanisms underlying the delayed emergence of emotion regulation effects and the development of targeted protocols for different PTSD subtypes represent important next steps in refining this therapeutic approach.

## 5. Conclusions

This investigation provides strong evidence that Amyg-EFP-NF represents a promising therapeutic approach for PTSD, demonstrating both immediate and progressive benefits across all symptom clusters. The temporal pattern of improvement, with initial symptom reduction followed by enhanced regulatory capabilities, points to a mechanism of action that involves both direct modulation of amygdala function and the development of improved emotion regulation skills.

The demonstrated effectiveness across diverse patient populations and symptom profiles, coupled with the durability of treatment effects, suggests that Amyg-EFP-NF could fill an important gap in current treatment options for PTSD. As understanding of the underlying mechanisms continues to grow, this approach may lead to increasingly refined interventions, potentially including adaptations based on symptom profiles, emotion regulation capacities, neural markers, and specific trauma types.

## Figures and Tables

**Figure 1 jcm-14-02421-f001:**
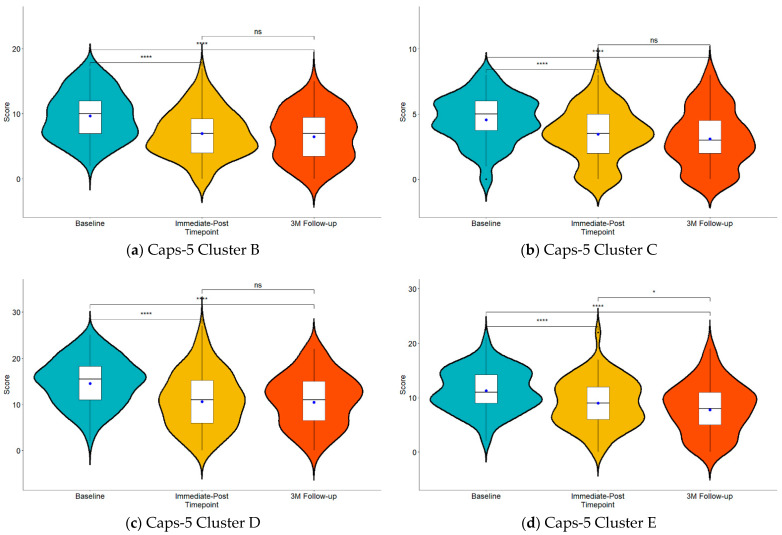
Violin plots showing the amygdala-derived-EFP-NF treatment effect on (**a**) intrusion (Cluster B), (**b**) avoidance (Cluster C), (**c**) cognition and mood (Cluster D), and (**d**) arousal and reactivity (Cluster E) symptoms of the CAPS-5 questionnaire. Blue dots in each plot represent the estimated marginal means of the LMM. *ns, p* > 0.05, * *p* ≤ 0.05, ** *p* ≤ 0.01, *** *p* ≤ 0.001, **** *p* ≤ 0.0001 adjusted *p*-values for multiple comparisons.

**Figure 2 jcm-14-02421-f002:**
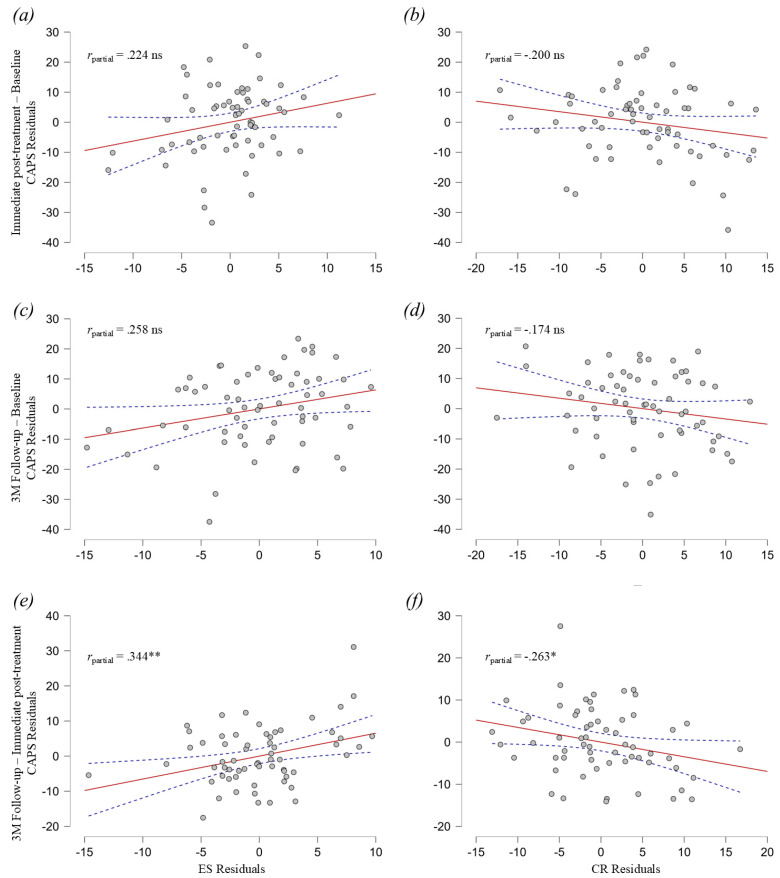
Partial regression plots for Expressive Suppression (ES) and Cognitive Reappraisal (CR) differences explaining the CAPS-5 difference. (**a**) Partial regression plot for ES and CAPS-5 differences from Baseline to Immediate-Post. (**b**) Partial regression plot for CR and CAPS-5 differences from Baseline to Immediate-Post. (**c**) Partial regression plot for ES and CAPS-5 differences from Baseline to 3 M Follow-up. (**d**) Partial regression plot for CR and CAPS-5 differences from Baseline to 3 M Follow-up. (**e**) Partial regression plot for ES and CAPS-5 differences from Immediate-Post to 3 M Follow-up. (**f**) Partial regression plot for CR and CAPS-5 differences from Immediate-Post to 3 M Follow-up. Dashed line area represents 95% confidence intervals. *ns*, *p* > 0.05, * *p* ≤ 0.05, ** *p* ≤ 0.01.

**Table 1 jcm-14-02421-t001:** Demographic and clinical characteristics across studies.

Characteristic	Study 1 (N = 25)	Study 2 (N = 37)	Study 3 (N = 66)
**Demographics**			
Age, mean ± SD, years	39.2 ± 12.3	36.8 ± 9.4	39.0 ± 10.6
Female, *n* (%)	12 (48.0%)	37 (100%)	35 (53.0%)
Education years, mean ± SD	14.1 ± 2.5	14.5 ± 4.4	14.3 ± 2.8
**Marital Status**			
Married, *n* (%)	12 (48.0%)	9 (24.3%)	32 (48.5%)
Single/Divorced/Separated, *n* (%)	13 (52.0%)	28 (75.7%)	34 (51.5%)
**Trauma Type**			
Combat/Military, *n* (%)	9 (36.0%)	0 (0%)	31 (47.0%)
Childhood sexual abuse, *n* (%)	4 (16.0%)	37 (100%)	12 (18.2%)
Interpersonal violence, *n* (%)	6 (24.0%)	0 (0%)	10 (15.1%)
Accidents/Other, *n* (%)	6 (24.0%)	0 (0%)	13 (19.7%)
**Clinical Characteristics**			
CAPS-5 total score, mean ± SD	35.4 ± 7.8	40.2 ± 8.1	42.5 ± 11.4
Time from trauma, mean ± SD, years	7.3 ± 5.2	13.5 ± 8.7	10.0 ± 5.7
Medication use, *n* (%)	17 (68.0%)	30 (81.1%)	48 (72.7%)
Concurrent psychotherapy, *n* (%)	9 (36.0%)	37 (100%)	30 (45.5%)
Prior trauma-focused therapy completed, *n* (%)	8 (32.0%)	13 (35.1%)	23 (34.8%)

Data compiled from Fruchtman-Steinbok et al. (2021) [27], Fine et al. (2023) [28], and Fruchter et al. (2024) [29].

**Table 2 jcm-14-02421-t002:** CAPS-5 symptom cluster scores.

Cluster	Timepoint	Mean [SD]		
		Study 1 (N = 25)	Study 2 (N = 37)	Study 3 (N = 66/63)
Intrusion (B)	Baseline	8.60 [3.61]	9.43 [3.90]	10.24 [3.54]
	Immediate-Post	6.48 [3.92]	7.05 [3.76]	7.21 [3.49]
	3 M Follow-up			6.71 [3.79]
Avoidance (C)	Baseline	4.28 [1.46]	4.73 [1.88]	4.59 [1.93]
	Immediate-Post	3.00 [1.61]	3.65 [2.24]	3.52 [2.03]
	3 M Follow-up			3.13 [2.21]
Cognition and Mood (D)	Baseline	11.88 [4.37]	15.38 [4.88]	15.71 [4.90]
	Immediate-Post	8.16 [4.84]	13.38 [5.88]	10.67 [6.15]
	3 M Follow-up			11.06 [5.55]
Arousal and reactivity (E)	Baseline	10.68 [3.48]	10.70 [3.88]	11.92 [3.92]
	Immediate-Post	8.32 [4.07]	9.00 [4.25]	9.26 [4.57]
	3 M Follow-up			8.10 [4.65]

In Study 3, the mean and standard deviation are reported for 66 subjects at Baseline and Immediate-Post and for 63 subjects at the 3 M Follow-up.

**Table 3 jcm-14-02421-t003:** CAPS-5 and ERQ scores for Study 3 (N = 63).

Timepoint	Mean [SD]		
	CAPS-5 Total	ERQ Cognitive Reappraisal	ERQ Expressive Suppression
Baseline	42.52 [11.46]	24.89 [8.34]	15.38 [6.03]
Immediate Post	30.49 [13.47]	25.71 [6.79]	14.62 [6.31]
3 M Follow-up	29.00 [13.61]	27.02 [6.78]	14.68 [5.98]

The mean and standard deviation are reported for the 63 subjects who completed the 3 M Follow-up.

## Data Availability

The data that support the findings of this study are available from the corresponding author upon reasonable request.

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
