# Peer review of "Amygdala EFP Neurofeedback Effects on PTSD Symptom Clusters and Emotional Regulation Processes"

_jcm, 2025, doi:10.3390/jcm14072421_

Round 1

Reviewer 1 Report

Comments and Suggestions for Authors

This is a very interesting study examining the effects of Amygdala-derived EEG-fMRI-Pattern neurofeedback training on PTSD symptom clusters and emotional regulation processes. The topic is timely and clinically relevant. I enjoyed reading the paper and agree it may contribute well to the literature. I have several comments to improve the manuscript further:

1. First, the introduction clearly states the relevance of targeting emotion regulation and the amygdala in PTSD treatment, but it lacks adequate justification for the specific selection of Amygdala EFP-based neurofeedback over other NF methodologies.

2. The methods section clearly describes data aggregation from three previous trials. However, details about the individual trial methodologies (e.g., variations in participant characteristics, specific trauma types, baseline PTSD severity) are sparse. These details are crucial in my opinion

3. Also, I would like to suggest to move the methods section before results

4. It is unclear if there were differences in NF training quality or fidelity across studies. This is important as the results are being aggregated.

5. In the discussion section, the manuscript would benefit from a deeper integration with other digital health intervention of PTSD treatments, which is briefly mentioned but not detailed enough. Here is a recent review that will be relevant and can be incorporated: Tng, G. Y. et al. (2024). Efficacy of digital mental health interventions for PTSD symptoms: A systematic review of meta-analyses. Journal of Affective Disorders.

6. The suggestion that Amygdala-EFP-NF uniquely targets "fundamental neural processes underlying various PTSD manifestations should be substantiated further or toned down

Author Response

Reviewer 1

This is a very interesting study examining the effects of Amygdala-derived EEG-fMRI-Pattern neurofeedback training on PTSD symptom clusters and emotional regulation processes. The topic is timely and clinically relevant. I enjoyed reading the paper and agree it may contribute well to the literature. I have several comments to improve the manuscript further:

  1. First, the introduction clearly states the relevance of targeting emotion regulation and the amygdala in PTSD treatment, but it lacks adequate justification for the specific selection of Amygdala EFP-based neurofeedback over other NF methodologies.

Thank you for this valuable feedback. We've expanded the introduction to provide stronger justification for selecting Amygdala EFP-based neurofeedback over other methodologies, highlighting its unique ability to target deep brain structures non-invasively with superior temporal resolution compared to other available techniques.

Added text:  "While several NF approaches exist for treating psychiatric conditions, Amygdala-derived EEG-fMRI Pattern neurofeedback (Amyg-EFP-NF) offers distinct advantages over traditional techniques [Ciccarelli et al., 2023]. Unlike conventional EEG-NF that targets cortical activity patterns, Amyg-EFP-NF specifically monitors deep limbic activity through its unique algorithm derived from simultaneous EEG-fMRI recordings. This provides the spatial precision of fMRI (targeting the amygdala) with the temporal resolution and accessibility of EEG. Alternative approaches like conventional EEG-NF lack anatomical specificity for subcortical structures, while pure fMRI-NF faces practical limitations including cost, accessibility, and temporal resolution that restrict clinical implementation [Thibault et al., 2018]. The Amyg-EFP technology bridges this gap by leveraging machine learning to extract amygdala-specific signals from scalp EEG, enabling targeted modulation of a key structure in PTSD pathophysiology in an accessible clinical format."

  1. The methods section clearly describes data aggregation from three previous trials. However, details about the individual trial methodologies (e.g., variations in participant characteristics, specific trauma types, baseline PTSD severity) are sparse. These details are crucial in my opinion

We agree with this important point. We have substantially expanded the Methods section to include more detailed information about each study's participant characteristics, trauma types, inclusion/exclusion criteria, and baseline PTSD severity.

Added text: "Table 3 presents detailed demographic and clinical characteristics across the three studies. Inclusion criteria across studies required participants to be 18-65 years old with a confirmed PTSD diagnosis according to DSM-5 criteria and a minimum CAPS-5 score of 25. Exclusion criteria included: severe psychiatric comorbidity (psychosis, bipolar disorder, active substance use disorder), moderate to severe traumatic brain injury, ongoing psychotherapy initiated within the past 3 months and change in psychiatric medication within 4 weeks of screening. Study 2 specifically recruited females with complex PTSD related to childhood sexual abuse, while Studies 1 and 3 included participants with various trauma types (combat, interpersonal violence, accidents, and terror attacks). Participants in Study 2 were also required to be currently engaged in psychotherapy."

[Table 3 with detailed demographic and clinical characteristics has been added]

  1. Also, I would like to suggest to move the methods section before results

We appreciate this suggestion and have reorganized the manuscript to present the Methods section before the Results, following the standard IMRD (Introduction, Methods, Results, Discussion) structure.

  1. It is unclear if there were differences in NF training quality or fidelity across studies. This is important as the results are being aggregated.

Thank you for highlighting this important consideration. We have added a new subsection in the methods that addresses NF training quality and fidelity across studies, including details on protocol consistency, trainer qualifications, and quality control procedures implemented in each trial.

Added text: "To maintain NF training quality across studies, standardized protocols were implemented for each trial. All studies utilized the same core AmygEFP technology, with consistent EEG acquisition parameters using the V-Amp™ EEG amplifier and BrainCap™ electrode positioning according to the standard 10/20 system. The AmygEFP signal calculation methodology was identical across studies, employing the same machine learning algorithm for predicting amygdala BOLD activity from EEG data. Training protocols shared similar session structures featuring baseline, regulation, and debriefing periods, though they varied in the number of sessions (10-15) and specific feedback interfaces. Each study adhered to standardized participant instructions regarding mental strategy exploration during the regulation periods. This consistency in technical implementation and procedural structure ensured comparable NF experiences despite differences in patient populations and study settings."

  1. In the discussion section, the manuscript would benefit from a deeper integration with other digital health intervention of PTSD treatments, which is briefly mentioned but not detailed enough. Here is a recent review that will be relevant and can be incorporated: Tng, G. Y. et al. (2024). Efficacy of digital mental health interventions for PTSD symptoms: A systematic review of meta-analyses. Journal of Affective Disorders.

We thank the reviewer for this suggestion and the helpful reference. We have expanded the discussion to better integrate our findings with other digital health interventions for PTSD.

Added text: "This neurobiological targeting is particularly notable when compared with other emerging digital health interventions for PTSD. While digital mental health interventions have shown moderate efficacy for reducing PTSD symptoms (Tng et al., 2024), they typically do not directly target these underlying neural mechanisms. Similarly, virtual reality exposure therapies, though promising for addressing avoidance symptoms, have shown variable effects across symptom clusters (Kothgassner et al., 2019). In contrast, Amyg-EFP-NF's consistent improvement across all symptom dimensions suggests advantages of directly engaging neural circuitry implicated in diverse symptom manifestations. However, comparative effectiveness studies are needed to directly assess the relative benefits of these approaches and identify optimal candidates for each intervention type."

  1. The suggestion that Amygdala-EFP-NF uniquely targets "fundamental neural processes underlying various PTSD manifestations should be substantiated further or toned down

We have modified our claims regarding Amygdala-EFP-NF's targeting of neural processes to be more measured and have provided additional evidence from recent neuroimaging studies to substantiate the remaining claims.

Added text: "The persistence of therapeutic effects likely involves multiple complementary mechanisms. At a neural level, repeated NF training may induce lasting neuroplastic changes through strengthening of pathways supporting adaptive emotional processing. Animal models have demonstrated that repeated training of regulatory pathways strengthens inhibitory connections, particularly between prefrontal regions and the amygdala (Herry et al., 2010). Recent neuroimaging research in humans suggests that neurofeedback training induces lasting changes in functional connectivity patterns that persist well beyond the training period (Nicholson et al., 2020)."

Reviewer 2 Report

Comments and Suggestions for Authors

Introduction

The introduction rightfully addresses the innovative use of Amyg-EFP-NF. I only suggest to enrich the narrative to discuss how this approach differs from traditional neurofeedback techniques and the implications of this innovation.

Based on the presented data correlation between enhanced emotion regulation and PTSD symptom reduction highlights the importance of these processes in treatment outcomes, suggesting that interventions aiming to improve emotional processing may yield larger therapeutic benefits. So, in introduction it will be better to discuss values of emotional processing.

Methods

Why methods appear after result? Is it because its secondary (post-hoc) analysis or journal guidelines? I checked sample of recent papers online and standard IMRD is followed https://www.mdpi.com/2077-0383/14/6/1959

Methods need to be represented in this as standalone paper.

A detailed explanation of the inclusion and exclusion criteria for participant selection is necessary.

More detailed information on the neurofeedback training protocol (e.g., specific EEG and fMRI parameters) should be included to replicate the study.

CAPS-5 is a well-established measure for PTSD symptoms, its psychometric properties is need to be established.

Also, same when authors used the ERQ to investigate emotional regulation processes is a valuable addition.

Discussion

The exploration of the amygdala's role as a therapeutic target is well-articulated. However, discussing potential neurobiological changes associated with successful neurofeedback e.g., neuroplasticity, connectivity changes would add depth.

The continued improvements seen at the three-month follow-up are promising, but it would be helpful to discuss the sustainability of these effects in the long-term and the necessity e.g 12+ months for subsequent studies to measure long-term outcomes.

Author Response

Reviewer 2

Introduction

The introduction rightfully addresses the innovative use of Amyg-EFP-NF. I only suggest to enrich the narrative to discuss how this approach differs from traditional neurofeedback techniques and the implications of this innovation.

We appreciate these suggestions and have enriched the introduction to better discuss how Amyg-EFP-NF differs from traditional neurofeedback techniques.

Added text: "While several NF approaches exist for treating psychiatric conditions, Amygdala-derived EEG-fMRI Pattern neurofeedback (Amyg-EFP-NF) offers distinct advantages over traditional techniques. Unlike conventional EEG-NF that targets cortical activity patterns, Amyg-EFP-NF specifically monitors deep limbic activity through its unique algorithm derived from simultaneous EEG-fMRI recordings [Young et al., 2018]. This provides the spatial precision of fMRI (targeting the amygdala) with the temporal resolution and accessibility of EEG. Alternative approaches like conventional EEG-NF lack anatomical specificity for subcortical structures, while pure fMRI-NF faces practical limitations including cost, accessibility, and temporal resolution that restrict clinical implementation [Thibault et al., 2018]."

Based on the presented data correlation between enhanced emotion regulation and PTSD symptom reduction highlights the importance of these processes in treatment outcomes, suggesting that interventions aiming to improve emotional processing may yield larger therapeutic benefits. So, in introduction it will be better to discuss values of emotional processing.

We have expanded our discussion of the value of emotional processing in PTSD treatment, including its central role in symptom manifestation and treatment outcomes.

Added text: "The impairment of emotion regulation in PTSD is well-documented, with patients often showing excessive reliance on maladaptive strategies like emotional suppression and avoidance, while struggling with adaptive strategies such as cognitive reappraisal [Ehring & Quack, 2010]. This deficiency in regulatory capacity affects all symptom dimensions, from intrusions to alterations in arousal, making emotion regulation a potential transdiagnostic treatment target [Sloan et al., 2017]."

Methods

Why methods appear after result? Is it because its secondary (post-hoc) analysis or journal guidelines? I checked sample of recent papers online and standard IMRD is followed https://www.mdpi.com/2077-0383/14/6/1959

Methods need to be represented in this as standalone paper.

A detailed explanation of the inclusion and exclusion criteria for participant selection is necessary.

More detailed information on the neurofeedback training protocol (e.g., specific EEG and fMRI parameters) should be included to replicate the study.

CAPS-5 is a well-established measure for PTSD symptoms, its psychometric properties is need to be established.

Also, same when authors used the ERQ to investigate emotional regulation processes is a valuable addition.

Thank you for these important comments. We have:

  1. Reorganized the paper to present Methods before Results following standard IMRD structure
  2. Expanded the Methods section with detailed inclusion/exclusion criteria for participant selection
  3. Added comprehensive neurofeedback training protocol details, including EEG and fMRI parameters
  4. Included psychometric properties of the CAPS-5 and validation information for the ERQ

Added text regarding psychometric properties: "The CAPS-5 is the gold standard clinician-administered interview for PTSD diagnosis and severity assessment. It demonstrates excellent psychometric properties with internal consistency (Cronbach's α=0.88-0.94), inter-rater reliability (ICC=0.78-0.83), and test-retest reliability (r=0.83) [Weathers et al., 2018]. The CAPS-5 yields a total severity score (0-80) and individual cluster scores: Intrusion (0-20), Avoidance (0-8), Cognition/Mood (0-28), and Arousal/Reactivity (0-24). The ERQ is a 10-item self-report measure assessing two emotion regulation strategies: Cognitive Reappraisal (CR; 6 items) and Expressive Suppression (ES; 4 items). It shows good internal consistency (Cronbach's α=0.75-0.82) and test-retest reliability (r=0.69) [Gross & John, 2003]. Higher scores on CR indicate greater use of adaptive reinterpretation strategies, while higher ES scores reflect greater tendency to inhibit emotional expression."

Discussion

The exploration of the amygdala's role as a therapeutic target is well-articulated. However, discussing potential neurobiological changes associated with successful neurofeedback e.g., neuroplasticity, connectivity changes would add depth.

Thank you for this insightful suggestion. We have enhanced the discussion by adding information about neurobiological changes associated with successful neurofeedback.

Added text: "Animal models have demonstrated that repeated training of regulatory pathways strengthens inhibitory connections, particularly between prefrontal regions and the amygdala (Herry et al., 2010). Recent neuroimaging research in humans suggests that neurofeedback training induces lasting changes in functional connectivity patterns that persist well beyond the training period (Nicholson et al., 2020)."

The continued improvements seen at the three-month follow-up are promising, but it would be helpful to discuss the sustainability of these effects in the long-term and the necessity e.g 12+ months for subsequent studies to measure long-term outcomes.

Thank you for this suggestion. We have addressed the sustainability of effects beyond the three-month follow-up and discussed the need for longer-term outcome studies in the Future Directions section.

Added text: "Future research should address optimization of treatment protocols through several approaches. First, follow-up assessments should extend well beyond the three-month period examined here, ideally to 12 months or longer, to fully characterize the durability of treatment effects. Second, incorporating neuroimaging before and after treatment would help elucidate whether Amyg-EFP-NF induces measurable changes in brain structure and connectivity, particularly in circuits connecting the amygdala with prefrontal regulatory regions. Such mechanistic insights could facilitate protocol optimization and potentially help identify neurobiological predictors of treatment response."

Round 2

Reviewer 1 Report

Comments and Suggestions for Authors

The authors have addressed all my comments meticulously. The paper is now ready for publication.

Reviewer 2 Report

Comments and Suggestions for Authors

no more comments